# Style-Talker: Finetuning Audio Language Model and Style-Based Text-to-Speech Model for Fast Spoken Dialogue Generation

**Yinghao Aaron Li & Xilin Jiang** [*]
Department of Electrical Engineering
Columbia University
New York, NY, USA
{yl4579, xj2289}@columbia.edu

**Jordan Darefsky & Ge Zhu**
Department of Electrical and Computer Engineering
University of Rochester
Rochester, NY, USA
{jdarefsk, ge.zhu}@rochester.edu

**Nima Mesgarani**
Department of Electrical Engineering
Columbia University
New York, NY, USA
nima@ee.columbia.edu

## Abstract

The rapid advancement of large language models (LLMs) has significantly propelled the development of text-based chatbots, demonstrating their capability to engage in coherent and contextually relevant dialogues. However, extending these advancements to enable end-to-end speech-to-speech conversation bots remains a formidable challenge, primarily due to the extensive dataset and computational resources required. The conventional approach of cascading automatic speech recognition (ASR), LLM, and text-to-speech (TTS) models in a pipeline, while effective, suffers from unnatural prosody because it lacks direct interactions between the input audio and its transcribed text and the output audio. These systems are also limited by their inherent latency from the ASR process for real-time applications. This paper introduces Style-Talker, an innovative framework that fine-tunes an audio LLM alongside a style-based TTS model for fast spoken dialog generation. Style-Talker takes user input audio and uses transcribed chat history and speech styles to generate both the speaking style and text for the response. Subsequently, the TTS model synthesizes the speech, which is then played back to the user. While the response speech is being played, the input speech undergoes ASR processing to extract the transcription and speaking style, serving as the context for the ensuing dialogue turn. This novel pipeline accelerates the traditional cascade ASR-LLM-TTS systems while integrating rich paralinguistic information from input speech. Our experimental results show that Style-Talker significantly outperforms the conventional cascade and speech-to-speech baselines in terms of both dialogue naturalness and coherence while being more than 50% faster. The demo and code are available at `https://styletalker.github.io/`.

## 1 Introduction

Conversing with a machine as if it were talking to a human has always been a dream for many computer scientists. Traditional spoken dialog systems (SDS) have relied on a multi-component architecture encompassing automatic speech recognition (ASR), language comprehension, dialog managers for turn-taking, response generation, and text-to-speech (TTS) synthesis. This complex pipeline enables interactions between humans and machines by converting input speech to text, understanding and managing the dialog's context,

---

[*]These authors contributed equally.

generating appropriate responses, and ultimately synthesizing these responses back into speech (Jokinen & McTear, 2022). Despite its effectiveness, this traditional setup has been challenged by the recent seismic shifts brought about by deep learning innovations.

Recent advances in deep learning, particularly in the development of large language models (LLMs), have dramatically simplified the SDS architecture dramatically (Zhao et al., 2023). By integrating language comprehension, turn-taking, and response generation into a single LLM, the dialog system pipeline has been condensed into a more streamlined ASR-LLM-TTS process (Mitsui et al., 2023; Yi et al., 2024). Furthermore, cutting-edge approaches now aim to achieve more direct end-to-end (E2E) speech-to-speech generation by treating speech as language tokens and modeling it similarly to LLMs, thereby eliminating the need for a separate ASR and TTS process (Lakhotia et al., 2021; Nguyen et al., 2023; Nachmani et al., 2023; Kim et al., 2024). This paradigm shift promises to further streamline dialog systems, making them more efficient and versatile.

However, both the ASR-LLM-TTS pipeline and speech LLM approaches exhibit significant limitations. The former struggles with capturing the emotional nuances of the input audio, failing to integrate this crucial aspect of human communication into the dialog process. While some research has attempted to address this by understanding input emotions (Xue et al., 2023), the synthesized response still remains emotionally disconnected from the input, as the TTS component operates independently of the ASR and LLM. Another very recent work (Lin et al., 2024) tries to incorporate speaking style in the LLM output for subsequent TTS synthesis. Still, the styles are derived from pre-defined categories, requiring extensive labeling works from more powerful LLM such as GPT-4, hindering its applications on in-the-wild datasets. Moreover, the autoregressive decoding required for ASR slows the entire system, impeding its applicability in real-time scenarios. On the other hand, the end-to-end speech LLM approach, despite its theoretical advantages, faces practical challenges in data acquisition and processing speed (Nguyen et al., 2023), making it less feasible for real-time applications due to the extensive computational resources required to generate speech units.

In this paper, we introduce Style-Talker, a novel SDS framework designed to overcome these challenges. Style-Talker innovatively integrates input audio directly with transcribed speech context of previous turns from the ASR model and its corresponding speaking style from a style-based TTS model. This approach not only preserves the prosodic (style) and semantic (text) aspects of speech but also significantly enhances the system's efficiency by eliminating the ASR component prior to the LLM in the response generation process. By training the audio LLM to output both the response text and its associated speech style, Style-Talker enables the synthesis of speech that accurately reflects the intended emotional and stylistic nuances. Concurrently, it processes the current input audio for transcription and style analysis, using this information to inform the next dialog turn. This seamless integration and the elimination of intermediate ASR processing dramatically improve the real-time factor (RTF) of the system, making Style-Talker nearly twice faster than the ASR-LLM-TTS cascade baseline with Whisper *large-v2* model (Radford et al., 2023) and more than four times faster than a recent speech-to-speech baseline, SpeechGPT (Zhang et al., 2023). Moreover, our system outperforms both the cascade and speech-to-speech baselines in generating responses that are not only more natural and coherent but also emotionally congruent with the dialog context. Since our system does not require manual labeling, it can also be applied directly to datasets mined in the wild, greatly diversifying its applicability and efficiency.

## 2 Related Works

Recent advancements in spoken dialog generation have primarily followed two distinct approaches: the text-based approach and the end-to-end (E2E) speech-to-speech approach. The text-based approach generates a text response that is subsequently converted into speech via text-to-speech (TTS) synthesis. In contrast, the E2E approach aims for direct speech output, bypassing the intermediate step for text generation altogether. Each of these methods, while effective, presents unique challenges and limitations that have spurred ongoing research efforts to improve the efficiency and naturalness of dialog systems.

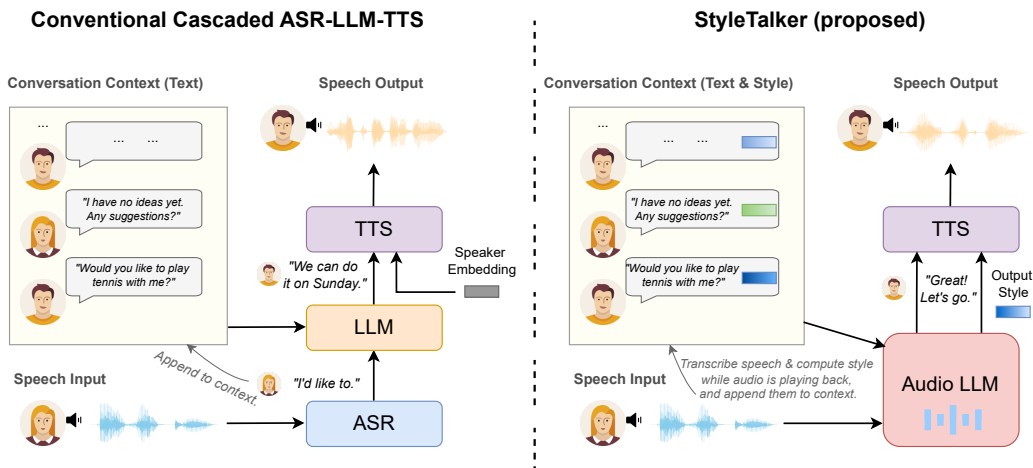

Figure 1: An overview of SDS with a comparison between the conventional cascaded system and Style-Talker. The cascaded system has three steps: input speech transcription (ASR), response text generation (LLM), and response speech synthesis (TTS). Style-Talker adopts an audio LLM to merge the first two steps and generate a response text and corresponding style directly, preserving the content prosody while achieving generation efficiency.

## 2.1 Text-Based Approaches and Paralinguistic Decoupling

The text-based approach, despite its widespread adoption, is inherently limited by the fundamental decoupling from the paralinguistic features of the input and output speech, such as tones, emotions, and intonations. Recent efforts in text-based dialog systems have sought to bridge the gap between speech content and its underlying paralinguistic information. For example, Xue et al. (2023) employed a speech encoder to inform LLMs about paralinguistic cues for generating emotionally congruent responses. Further, Lin et al. (2023) and Lin et al. (2024) introduced multimodal approaches that combine text and speech inputs, or text with emotion vectors, to better interpret these cues. Yet, these methods require speech transcription into text, introducing latency that impacts processing speed. Contrary to these, Style-Talker directly processes speech inputs, circumventing transcription delays and achieving significant speed improvements without sacrificing prosodic or conversational coherence.

Moreover, efforts have been made to enhance the paralinguistic aspects of generated responses by incorporating emotion labels into text responses (Varshney et al., 2021; Liu et al., 2021; Lin et al., 2023; 2024). These works, however, rely on predefined text labels for each utterance, complicating application to diverse and unstructured in-the-wild datasets. Style-Talker, in contrast, adopts a self-supervised approach with a style-based TTS model, learning speech styles directly optimized for speech synthesis. This method mitigates limitations of predefined emotion categories, offering a more flexible and end-to-end solution.

## 2.2 End-to-End Speech-to-Speech Generation

E2E models represent a paradigm shift in spoken dialog systems by generating speech output directly from speech input, modeling speech in an autoregressive fashion, which has been used both for dialog generation Nguyen et al. (2023); Kim et al. (2024) and translation Duquenne et al. (2023); Barrault et al. (2023). The discrete dGSLM framework (Nguyen et al., 2023), for example, treats speech as discrete units akin to text tokens, achieving emotional and prosodic coherence. Spectron (Nachmani et al., 2023) extends this E2E framework by directly generating mel-spectrograms instead of discreet tokens. However, due to data limitations, these E2E dialog systems trained from scratch on speech corpora often lack semantic coherence in dialog responses. SpeechGPT (Zhang et al., 2023; 2024) and USDM (Kim et al., 2024) attempt to address this by integrating text-based LLMs with speech datasets, though challenges in achieving human-like speech quality remain, alongside

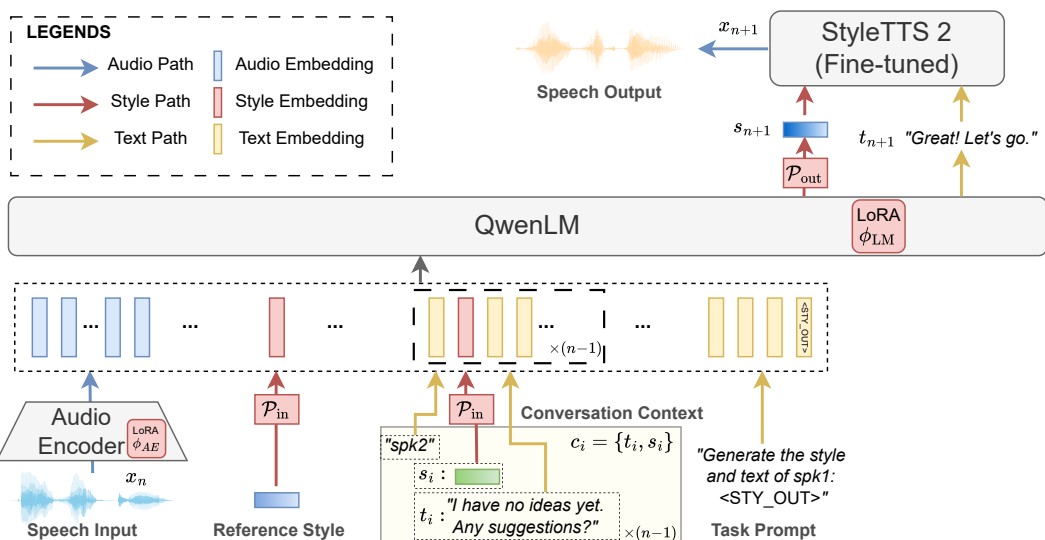

Figure 2: Model components and processing pipelines of Style-Talker for response generation. Audio $x_n$ from the incoming speaker, a reference and past speaker styles in conversation, and transcriptions from previous rounds ($c_n$) are all embedded into the same space by the audio encoder, the style projection $\mathcal{P}_{\text{in}}$, and the text tokenizer and embedder (not shown), respectively. They are jointly processed by an LLM QwenLM to generate the response text $t_{n+1}$ and style $s_{n+1}$ for StyleTTS 2 to synthesize a response speech $x_{n+1}$.

the complexities of training models across multiple modalities. Unlike the E2E approach, Style-Talker fine-tunes an audio language model and a style-based TTS model to generate semantically and paralinguistically coherent responses. This approach not only enhances dialog naturalness but also offers substantial improvements in processing speed, marking a new advancement for dialog generation technology.

## 3 Methods

### 3.1 Style-Talker

Style-Talker is a spoken dialog system (SDS) designed for seamless, real-time speech-to-speech interaction by integrating two major components: a multi-modal audio large language model (LLM) for spoken language understanding (SLU) and text-style response generation, and a style-based text-to-speech (TTS) model for synthesizing speech that mirrors the specific speaking style indicated by the style vector generated from the language model. This innovative architecture enables Style-Talker to produce responses that are not only contextually relevant but also paralinguistically congruent with the previous speaker's speaking style, significantly enhancing the naturalness and coherence of dialogues.

The training of Style-Talker involves a two-stage process. Initially, the StyleTTS 2 model (Li et al., 2024), a state-of-the-art style-based TTS model, is fine-tuned on a dialog dataset wherein speakers' utterances are diarized or segmented into individual speech sentences. The model learns the nuances of speaking styles in a self-supervised manner, without the need for explicit style annotations, with which the speaking styles are extracted by the style encoder into a fixed-length style vector. Subsequently, an audio LLM, such as Qwen-Audio (Chu et al., 2023), is fine-tuned to generate both the response text and its associated speaking style, utilizing the previously transcribed text and style from each utterance as contextual input. This phase is critical for aligning the generated responses with the conversational context through texts and its underlying paralinguistic features via styles, thus ensuring coherence and style consistency in dialogues.

During inference, we provide a reference style vector of the response speaker as a target speaker embedding at the beginning of the input prompt. For simplicity, we do not model

turn-taking explicitly. The system instead infers the end of a speaker's turn based on a predetermined time threshold. If the speaker's silence exceeds this threshold, the system proceeds to process the input speech. For each incoming utterance $x_n$, the fine-tuned audio LLM takes both the speech input $x_n$ and its preceding context $c_{n-1}$ as detailed in Section 3.1.3. The model then generates the response text $t_{n+1}$ and the corresponding speaking style $s_{n+1}$. These outputs are fed into the TTS model, which synthesizes the speech $x_{n+1}$ that reflects both the content and style of the response, playing it back to the user.

Simultaneously, as the response speech is being played back, the system employs an ASR model to transcribe the input speech $x_n$ into text $t_n$, and a style encoder from StyleTTS 2 to extract the speaking style into a vector $s_n$. These elements are then aggregated into the contextual data $c_n = \{t_i, s_i\}_{i=1}^n$ for processing the next speech input $x_{n+2}$. This methodology effectively reduces the system's reliance on ASR during the user's wait time for a response, thereby improving the real-time applicability of the system. By streamlining the dialog process to involve only a single LLM decoder for response generation, rather than two separate autoregressive decoders for both ASR and response generation, Style-Talker achieves remarkable efficiency and speed, signifying a new step toward real-time spoken dialog systems. In the following section, we detail each component, context aggregation, and prompt settings. For more implementation details, please refer to Appendix A.

### 3.1.1 StyleTTS 2

StyleTTS 2 is a state-of-the-art TTS model with human-level performance and public pre-trained checkpoints. It differs from other TTS models in how it captures and represents the speaking style. At the core of StyleTTS 2's design is an autoencoder framework that learns a fixed-length vector capturing the speaking style as its latent representation, from which the speech is decoded back to waveform conditioned on the transcribed text. This style representation is comprehensive, containing a broad range of paralinguistic features that extend beyond the mere content of speech. It includes, but is not limited to, the speaker identity, prosody, lexical stress, formant transitions, and speaking rate (Li et al., 2024). In essence, the speaking style vector serves as a paralinguistic summary of the speech, effectively distilling the essence of how something is said, separate from what is being said.

To leverage the capabilities of StyleTTS 2 for dialog generation, we begin by fine-tuning the model using the LibriTTS checkpoint [1]. We prepared our dataset at the utterance level, with each utterance attributing to a single speaker (see Section 4.1). This is because the style encoder of StyleTTS 2 is designed to process speech from individual speakers and cannot encode utterances with multiple speakers. We then compute the utterance-level style for each utterance in the dataset for audio LM fine-tuning.

### 3.1.2 Qwen-Audio

Qwen-Audio is an open-source multi-modal language model, combining the text and audio modalities to understand and respond to both text and audio inputs. This model is an adaptation of Qwen-7B (Bai et al., 2023), fine-tuned with an audio encoder from Whisper *large-v2* (Radford et al., 2023) [2]. This integration enables Qwen-Audio to comprehend audio inputs and generate corresponding text responses with the capacity to understand both the spoken content and its paralinguistic attributes. In Style-Talker, we have further fine-tuned Qwen-Audio to incorporate speaking style both as an input and output modality. We modified the input head to the transformer layers to accept not only text information of previous conversation turns but also the associated speaking style as the context (see Section 3.1.3). The details of this modification, including specific prompts used, can be found in Appendix A.1. In this setup, the speaking style is transformed into a hidden representation that aligns with those of the text tokens in dimensionality through a linear projection layer $\mathcal{P}_{\text{in}}$. Similarly, when generating responses, an additional linear projection head $\mathcal{P}_{\text{out}}$ is applied to a special <STY_OUT> token appended near the end of the model's input sequence to decode the speaking style corresponding to the response text (see Figure 2 for detailed

---

[1] Available at `https://github.com/yl4579/StyleTTS2`
[2] Available at `https://github.com/QwenLM/Qwen-Audio`

illustrations). Since we only care about the paralinguistic information of the response speech rather than the pre-determined response speaker identity, we only predict the prosodic style in StyleTTS 2 and use a pre-computed acoustic style for the timbre of the target speaker's voice. We found that this approach helps retain the speaker's identity while maintaining the ability to produce prosodically coherent speech responses.

### 3.1.3 Conversation Context

In multi-turn dialog systems, the context of the conversation plays a pivotal role in generating responses that are both congruent and coherent. Research has consistently shown that incorporating longer contexts leads to more meaningful and contextually appropriate dialogues (Agarwal et al., 2018). Traditional text-based spoken dialog systems, however, typically limit context to the text content of previous interactions (Xue et al., 2023; Lin et al., 2024). This approach falls short in capturing the full spectrum of communicative nuances, particularly in regards to the speaking style, due to the absence of speech information within the dialog context. To address this limitation, Style-Talker enhances context representation by including both the text of each utterance and its corresponding speaking style. This enriched context is structured within the prompt in a novel manner: adjacent to each speaker's name, and preceding the transcribed speech content, a special style token is introduced. This token directly corresponds to the speaking style vector for the utterance, effectively embedding paralinguistic information within the dialog context (see Appendix A.1 for more details).

The integration of texts and style vectors into the conversation context is achieved through the use of ASR, with the Whisper model and the style encoder from StyleTTS 2. This process occurs while the current response speech is being played back to the user. This design choice minimizes potential delays in response time attributed to input transcription, ensuring the system remains highly responsive. By maintaining a comprehensive context that encompasses both semantic content and paralinguistic attributes, Style-Talker significantly advances the capabilities of spoken dialog systems. Since our model operates on the text domain, we can have a speech context spanning for up to 2 minutes, much longer than speech-to-speech models such as SpeechGPT, which often suffer from limited context windows, as speech is inherently longer than its transcribed texts.

### 3.2 Training Objectives

For a more natural style and coherent content in the response speech, we jointly optimize the style error and the next token probability. The style loss $\mathcal{L}_{\text{style}}$ is defined as the L1 distance between the ground truth prosodic style $\hat{s}_{n+1}$ and the style $s_{n+1}$ decoded from the last layer representation $h_{\texttt{<STY\_OUT>}}$ of the special token $\texttt{<STY\_OUT>}$:

$$\mathcal{L}_{\text{style}} = |s_{n+1} - \hat{s}_{n+1}|, \tag{1}$$

$$s_{n+1} = \mathcal{P}_{\text{out}}(h_{\texttt{<STY\_OUT>}}). \tag{2}$$

The response text generation maximizes the next text token ($t_{n+1}$) probability given the incoming speech $x_n$ and preceding context $c_{n-1}$ containing both styles and transcriptions:

$$\max_{\theta_{\text{in}}, \phi_{\text{AE}}, \phi_{\text{LM}}} P(t_{n+1} | x_n, c_{n-1}), \tag{3}$$

where $\theta_{\text{in}}$, $\phi_{\text{Audio}}$, $\phi_{\text{LM}}$ are the trainable parameters in the style input projection, and the audio encoder and the LLM in Qwen-Audio. We optimize the above objective through a cross-entropy loss $\mathcal{L}_{\text{text}}$ for $1 < t \leq T$ given a text sequence of length $T$.

The final loss is a weighted sum of the style and text loss: $\mathcal{L} = \mathcal{L}_{\text{text}} + \lambda \mathcal{L}_{\text{style}}$, where $\lambda$ is a hyperparameter for weighing different loss terms.

| Dataset | Model | MOS-N (CI) | MOS-C (CI) |
|---------|-------|-----------|-----------|
| DailyTalk | Ground Truth | 3.83 ($\pm$ 0.11) | 4.33 ($\pm$ 0.07) |
| | SpeechGPT (Zhang et al., 2023) | 2.87 ($\pm$ 0.08) | 3.11 ($\pm$ 0.09) |
| | Cascade (Whisper + Qwen-7B + StyleTTS 2) | 3.24 ($\pm$ 0.10) | 3.63 ($\pm$ 0.08) |
| | Style-Talker (Proposed) | **3.55** ($\pm$ **0.09**) | **3.90** ($\pm$ **0.09**) |
| PodcastFillers | Ground Truth | 4.22 ($\pm$ 0.08) | 4.18 ($\pm$ 0.09) |
| | Cascade (Whisper + Qwen-7B + StyleTTS 2) | 3.22 ($\pm$ 0.09) | 3.53 ($\pm$ 0.10) |
| | Style-Talker (Proposed) | **3.76** ($\pm$ **0.09**) | **4.02** ($\pm$ **0.09** ) |

Table 1: Mean opinion score of naturalness (MOS-N) and coherence (MOS-C) with 95% confidence intervals (CI) from human subjective evaluations.

## 4 Experiments

### 4.1 Datasets

We evaluated our system and baseline systems on two benchmark datasets: DailyTalk (Lee et al., 2023) and PodcastFillers (Zhu et al., 2022). The former is a two-speaker studio-recorded conversation dataset, and the latter is a spoken dialog dataset mined in the wild.

DailyTalk (Lee et al., 2023) is a multi-turn conversation dataset comprising 20 hours of spoken dialog data from two speakers, one male and one female, totaling 2,541 conversations. The dataset was recorded in a studio with contrived scripts by voice actors of non-native English speaker origins. We kept 100 and 200 conversations as validation and test set, respectively, and the remaining 2,241 conversations were used as the training set.

Although DailyTalk offers transcribed conversational speech, the audio recordings are contrived rather than captured in natural, real-world settings. We thus also conducted experiments on the PodcastFillers (Zhu et al., 2022), a dataset of conversations from podcast recordings produced in uncontrolled settings. The PodcastFillers dataset consists of 145 hours of gender-balanced podcast recordings featuring over 350 speakers, sourced from SoundCloud. It contains 199 podcasts in total. Since transcriptions provided with the PodcastFillers dataset do not contain speaker diarizations and filler words, we diarized and re-transcribed the dataset (see Appendix A.4 for details). We divided the dataset by podcast, with a 173/6/20 split for training, validation, and test sets.

### 4.2 Evaluations

We evaluated our SDS against two other systems: the traditional ASR-LLM-TTS cascade system and the end-to-end (E2E) speech-to-speech approach. Since there is no publicly available E2E speech-to-speech baseline with multi-turn support at the point the paper is written, following Kim et al. (2024), we fine-tuned a single-turn E2E model, SpeechGPT (Zhang et al., 2023) *7B-cm* [3], as the E2E baseline. Due to the limited context window SpeechGPT presents, we failed to fine-tune it on the more complicated PodcastFillers dataset to generate meaningful responses. Thus, we only evaluated SpeechGPT on the simpler DailyTalk dataset as in Kim et al. (2024). For more details of our baseline systems, including model choices of cascade baseline, please refer to Appendix A.3.

### 4.2.1 Subjective Evaluations

We used two metrics for the mean opinion score (MOS) as subjective evaluations: MOS-N for naturalness and MOS-C for coherence. Unlike Zhang et al. (2024) that only evaluates the speaker similarity, we asked the evaluators to rate the coherence of the generated speech, similar to how written dialogues are evaluated in text (Zhong et al., 2022). Both metrics involve the speech and the content of the responses. For MOS-N, the raters were only instructed to rate the naturalness of the response, that is, how likely the speech response is

---

[3]Avaiable at `https://huggingface.co/fnlp/SpeechGPT-7B-cm`

| Dataset | Model | BLEU ↑ | ROUGE-L ↑ | METEOR ↑ | BERT Score ↑ | WER ↓ |
|---|---|---|---|---|---|---|
| DailyTalk | SpeechGPT | 1.21 | 9.02 | 14.89 | 77.13 | 22.58 |
| | Cascade | 3.00 | 11.92 | 22.48 | 88.99 | 16.22 |
| | Style-Talker | **4.62** | **16.14** | **25.72** | **90.40** | **12.01** |
| PodcastFillers | Cascade | 0.43 | 7.08 | 10.44 | 83.98 | **10.61** |
| | Style-Talker | **3.31** | **10.06** | **17.00** | **84.26** | 15.59 |

Table 2: Semantic evaluations of F-1 score (%) for BLEU, ROUGE-L, METEOR, BERT Score and word error rate (WER). WER was computed with filler words and punctuation removed.

produced by a human instead of a robot based on both the speech quality and the content spoken. In contrast, MOS-C measures how well the response speech follows the previous conversation context, given the speech's content and the speaker's prosody and identity (see Appendix C for details). These evaluations were conducted by native English speakers from the U.S. on Amazon MTurk . For each test, we included 10 to 20 conversations, each provided with the previous 3 turns of conversation as the context. Each speech response set was rated by 5 to 10 evaluators on a 1-5 scale, with increments of 1. We followed Li et al. (2021) that randomized the model order and kept their labels hidden, similar to MUSHRA (BS, 2003). We used the rating of the ground truth response as the attention checker and excluded raters who did not rank the ground truth as the highest for both MOS metric.

### 4.2.2 Objective Evaluations

We followed Lin et al. (2024) to evaluate the response speech at both acoustic and semantic levels. Since our speaking styles are self-supervised, unlike Lin et al. (2024) with categorical and pre-defined styles where the generated styles can be compared against the ground truth labels for F-1 score, we conducted acoustic evaluations following Li et al. (2022) by calculating the correlations of several acoustic metrics for emotion recognition (Busso et al., 2013) between generated and ground truth speech. These metrics include pitch mean, pitch standard deviation (STD), energy mean, energy standard deviation (STD), and harmonics-to-noise (HTN) ratio. We also added speech duration as an extra metric to evaluate whether the generated speech follows the same distribution as the ground truth in terms of speech duration. Moreover, we scored the speaker similarity by employing a pre-trained WavLM speaker embedding model [4] following Ju et al. (2024).

We used the generated text directly for semantic evaluation, as all models evaluated, including SpeechGPT, generate text as responses. We followed Lin et al. (2024); Kim et al. (2024) and employed commonly used text-generation metrics, including BLEU (Papineni et al., 2002), ROUGE (Lin, 2004), METEOR (Banerjee & Lavie, 2005) and BERT Score (Zhang et al., 2019) against the ground truth text. Additionally, we calculated the word error rate (WER) with Whisper *large-v2* as a semantic metric to evaluate the intelligibility of the generated speech. All metrics were computed with the Huggingface Evaluate package [5].

Lastly, to test the real-time applicability, we computed the real-time factor (RTF), which is the ratio between the time used to generate a speech utterance and the duration of the generated speech, and the system delay by generating a 10-second response to a 10-second input utterance. The evaluation was conducted on a single Nvidia L40 GPU.

## 5 Results

### 5.1 Model Performance

According to the subjective evaluation results presented in Table 1, Style-Talker achieves a significant improvement in conversation naturalness and coherence against both the cascade and speech-to-speech baselines on the DailyTalk dataset. Furthermore, when

---

[4] Available at `https://huggingface.co/microsoft/wavlm-base-plus-sv`

[5] `https://huggingface.co/docs/evaluate/`

| Dataset | Model | Pitch mean | Pitch STD | Energy mean | Energy STD | HTN ratio | Speech duration | Speaker similarity |
|---|---|---|---|---|---|---|---|---|
| DailyTalk | SpeechGPT | 0.62 | 0.07 | 0.70 | 0.00 | 0.39 | 0.12 | 0.75 |
| | Cascade | 0.76 | 0.13 | 0.72 | 0.03 | 0.53 | 0.19 | 0.83 |
| | Style-Talker | **0.81** | **0.26** | **0.73** | **0.07** | **0.56** | **0.25** | **0.86** |
| PodcastFillers | Cascade | **0.72** | 0.24 | **0.82** | 0.44 | 0.45 | 0.17 | 0.82 |
| | Style-Talker | 0.68 | **0.28** | **0.82** | **0.58** | **0.51** | **0.20** | **0.84** |

Table 3: Comparison of Pearson correlation coefficients of acoustic features associated with emotions and speech duration between ground truth and response speech and cosine similarity of speaker embeddings between ground truth and response speech.

evaluated on the PodcastFillers dataset, which comprises recordings in more realistic and noisy settings, Style-Talker still outperforms the cascade baseline, showcasing its robustness and adaptability in various settings, whereas SpeechGPT fails to produce meaningful speech. Importantly, Table 4 shows that our system is nearly twice as fast as the cascade system and four times faster than the SpeechGPT baseline. Our system only generates 10-second audio from 10-second input audio in around 1.5 seconds, significantly faster than other baseline systems, making it feasible for real-time applications in human-computer interaction.

In terms of semantic similarity to ground truth responses, Style-Talker showed a strong alignment, as evidenced in Table 2. This is notable even as both the baseline systems and Style-Talker experience reduced semantic metrics on the PodcastFillers dataset due to its complexity from a more natural recording setting. Remarkably, Style-Talker consistently outperforms the cascade baselines in text dialog generation, even though semantic evaluation occurs in the text domain without any involvement of other modalities. This suggests that Style-Talker's architecture, which integrates audio input and previous speaking styles into the context for generating text and style, contributes positively to the semantic quality of generated text. This aspect is further explored in the ablation study detailed in Section 5.2, highlighting the semantic significance of incorporating audio and style context in dialog generation. In Table 3, acoustic features of the speech responses generated by Style-Talker show a higher correlation with those of the ground truth responses when compared to other baseline systems. This evidence strongly supports the efficacy of Style-Talker's design in capturing and reproducing the nuanced acoustic properties associated with prosody and emotion in human speech, proving its ability to produce emotionally coherent responses.

## 5.2 Ablation Study

To assess the contribution of each component to our system's performance, we conducted an ablation study by evaluating the system under various configurations, involving the addition or omission of specific features. We focused on three variants of our system: one excluding style vectors from the dialog context ("– *style in context*"), another substituting the audio language model (LM) with a conventional LM and instead using the text transcription and style of the input speech ("– *audio input*"), and lastly our proposed system provided with the additional text transcription and style vector of input speech ("+ *ASR & style*"). We summarized the results in Table 5 by averaging all metrics evaluated and added the averaged metrics on both DailyTalk and PodcastFillers (see Appendix A.4 for full results).

Table 5 shows that omitting the speaking styles from previous dialog turns adversely affects the quality of speech responses, impacting both semantic and acoustic aspects. This underscores the importance of incorporating speech styles into the context to produce responses that are semantically rich and prosodically aligned with the previous conversation. Conversely, substituting audio input with its text transcription and style vector has a less detrimental effect. This can be attributed to the style vector serving as a summary of the input's paralinguistic features. However, this configuration requires the use of an ASR model, markedly slowing down the system's processing speed. It is notable that including the current input speech's text and style does not enhance system performance, suggesting that the audio encoder is able to extract necessary information from the input audio that is contained in the text and style vector, making these additional inputs redundant. This

| Model | RTF | Delay |
|---|---|---|
| Style-Talker | **0.3873** | **1.53** s |
| SpeechGPT | 1.3246 | 13.82 s |
| Cascade | 0.5912 | 2.31 s |

Table 4: The real-time factor (RTF) and system delay when processing a 10s input and producing a 10s output audio between various models.

| Model | Semantic ↑ | Acoustic ↑ | RTF ↓ |
|---|---|---|---|
| Proposed | **62.88** | **1.07** | 0.3824 |
| – *style in context* | 61.21 | 1.01 | **0.3792** |
| – *audio input* | 62.10 | 1.04 | 0.6912 |
| + *ASR & style* | 62.76 | **1.07** | 0.6857 |

Table 5: Ablation study results with aggregated semantic and acoustic scores (sum of averaged metrics in Table 6 and Table 7 on both DailyTalk and PodcastFillers datasets) and real-time factor.

finding highlights the efficiency of the audio encoder in extracting relevant text and style cues without the need for explicit provision of such information.

## 6 Conclusions and Limitations

In this work, we presented Style-Talker, a novel spoken dialog system (SDS) that integrates audio input with its transcribed speech context and speaking style. Our system demonstrates superior performance in naturalness, coherence, and processing speed on challenging datasets compared to traditional ASR-LLM-TTS cascades and end-to-end baselines. Despite these significant advancements and our careful attention to real-time responsiveness, Style-Talker still faces limitations in response speed and may not always surpass SDS with real-time ASR implemented. Moreover, the quality of the generated speech is dependent on the downstream TTS model's performance, which may degrade when handling in-the-wild noisy datasets.

Future work will focus on enhancing real-time processing by encoding audio input in a causal manner and developing methods to summarize audio efficiently with minimal impact on input context token length. Improving the performance of TTS models for noisy, real-world datasets is another key area for further research. Additionally, our current system does not include explicit turn-taking mechanisms, so future research will explore incorporating efficient turn-taking strategies into the proposed framework.

## 7 Acknowledgements

We thank Cong Han for helping conducting subjective evaluations during the development of this work. This work is funded by the National Institutes of Health (NIH- NIDCD) and a grant from Marie-Josee and Henry R. Kravis.

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

# A    Implementation Details

## A.1    Prompt Settings

We adopted the input prompting format from Qwen-Audio and added extra tokens to accommodate speaker styles. The example below is the prompt template for two speakers with an ongoing conversation of three rounds. The last round is the incoming speech (the audio input to Qwen-Audio), in which the transcription and the style have not yet been calculated and appended to the context. Texts in < > are special tokens for Qwen-Audio's tokenizer, and texts in [ ] vary for each conversation.

```
Audio 1:<audio>[audio path]</audio>

This is the voice of the [spk a] last speaking. There is a conversation
among [spk a]: STYLE: <|extra_123|> [spk b]: STYLE: <|extra_123|>.
Here is some context:

[spk a]: STYLE: <|extra_123|> TEXT: [text]
[spk b]: STYLE: <|extra_123|> TEXT: [text]

Try to recognize what [spk a] just said from the audio,
and generate the style and text of the next speaker [spk b].
Be creative and avoid repeated words and sentences.
STYLE: <|extra_124|> TEXT:
```

We chose Qwen-Audio's pre-allocated special tokens <|extra_123|> as placeholders for input styles and <|extra_124|> as the location identifier for the output styles. <|extra_123|> tokens are not embedded by the text embedder. Projected input styles replace the embedding at these locations. The response style is predicted from the last layer representation of the <|extra_124|> token.

We have also crafted similar prompt templates for the baseline and ablation experiments. For example, we removed all styles in the context or removed the audio input for the – *style in context* and – *audio input* models in Table 5, respectively.

## A.2    Training and Inference Details

For both the baselines and our system, we randomly cropped the dialog, used every turn before the cropping point as the context, and trained the model to predict the response of the next turn. We truncated the context to fit the context window for both the baseline and our systems, 512 for SpeechGPT and 1536 for Qwen-7B and Qwen-Audio. 1536 is the maximum context window we can hold in the GPU memory. We used fixed dialog crops that were randomly sampled beforehand for evaluation and randomly sampled new crops for training.

We fine-tuned Qwen-Audio in Style-Talker and Qwen in the baseline using LoRA (Hu et al., 2021). We added LoRA to query, key, and value metrics in self-attention layers and weight metrics in MLPs in both the LLM and the audio encoder. We chose a rank of 16, a dropout ratio of 0.05, and no bias term. For fair comparisons, we trained Style-Talker, baseline, and ablation models with the same training hyperparameters. All models were trained with an AdamW optimizer, a peak learning rate of $1e - 4$, a linear learning rate decay, a maximum gradient norm of 1, a batch size of 1, and a gradient accumulation step of 8 (for an equivalent batch size of 8). We used mixed (bfloat16) precision for training on a Nvidia L40 GPU. We trained all models for 20 epochs on the DailyTalk dataset or 100 epochs on the PodcastFilters dataset.

In evaluation, we set the style diffusion parameter $\alpha = 0.3, \beta = 0$ for the DailyTalk dataset and $\alpha = \beta = 0$ for the PodcastFilters dataset for all models. We fixed the reference style from the context for a consistent evaluation and sampled with a temperature of 1 and top_k of 0.8.

### A.3 Baseline Systems

For the cascade system, we employed Whisper *large-v2* as the ASR model since the audio encoder in our system is fine-tuned from the encoder of Whisper *large-v2*. We fine-tuned a Qwen-7B model as the LLM for the cascade baseline, as our audio LLM was fine-tuned from Qwen-7B. We kept the same fine-tuned StyleTTS 2 model in both the cascade baseline and our proposed system.

For both the cascade baselines and our system, we used the previous 3 turns as the context for the input. For SpeechGPT, we only used the previous turn as its context window is limited to only one turn of input audio. Since SpeechGPT generates speech at 16 kHz while StyleTTS 2 produces speech at 24 kHz, we downsampled all audio to 16 kHz for DailyTalk dataset evaluations for fair comparison.

### A.4 PodcastFiller Dataset

Since we need "verbatim" transcriptions, we re-transcribed the podcast audio using Whisper (Radford et al., 2023) and use Pyannote (Bredin et al., 2020) to perform speaker diarization. Specifically, we first apply speaker diarization to approximately parse the long recordings into segments according to different speaker identities. Following this, we use a Whisper *large-v2* model to obtain transcriptions for these segments. In our experience, the pre-trained Whisper models tend to omit speaker turns, filler words, and other disfluencies; to address this, one can either prompt the model with diarized verbatim text [6] or fine-tune the model using verbatim data. To address this, we use a Whisper *large-v2* model that has been fine-tuned on a small dataset comprised of verbatim captions containing speaker turns, e.g., *"...[S1] Um, How are you today? [S2] Oh, I'm fine. Than-Thank you!"*. Lastly, after obtaining transcriptions, we filter out utterances that were not properly diarized by discarding segments whose transcripts contain multiple speaker indicators, e.g. containing both "[S1]" and "[S2]".

| Dataset | Model | BLEU | ROUGE-L | METEOR | BERT Score |
|---|---|---|---|---|---|
| DailyTalk | Proposed | **4.62** | **16.14** | 25.72 | 90.40 |
| | − *style in context* | 3.79 | 15.37 | 23.94 | 89.57 |
| | − *audio input* | 4.43 | 15.99 | **25.79** | 89.53 |
| | + *ASR & style* | 4.61 | 15.29 | **26.13** | 89.27 |
| PodcastFillers | Style-Talker | 3.31 | 10.06 | **17.00** | **84.26** |
| | − *style in context* | 3.02 | 9.44 | 15.49 | 84.19 |
| | − *audio input* | 3.05 | 9.55 | 15.96 | 84.13 |
| | + *ASR & style* | **3.65** | **11.08** | 16.91 | 84.11 |

Table 6: Semantic evaluations of F-1 score (%) for BLEU, ROUGE-L, METEOR, and BERT Score for ablation study. The results for the proposed model are copied from Table 2.

## B Detailed Ablation Study Results

In our ablation study, we conducted objective evaluations at both the acoustic and semantic levels to scrutinize the impact of various components within our proposed Style-Talker system. We focused on understanding how the inclusion or omission of specific features influences the system's overall performance, particularly in terms of acoustic quality and semantic accuracy. Notably, we opted to exclude the word error rate (WER) from our semantic evaluation metrics. This decision was informed by the observation that WER did not exhibit significant variability across the different configurations tested in our ablation study. Furthermore, its inclusion was found to introduce noise into the aggregated scores presented in Table 5, potentially obfuscating the true impact of the system modifications. By focusing on metrics that more directly reflect the semantic integrity and coherence of

---

[6]`https://github.com/openai/openai-cookbook/blob/main/examples/Whisper_prompting_guide.ipynb`

the generated responses, we aimed to achieve a clearer and more meaningful assessment of each component's contribution to the system's efficacy. The full evaluation results for semantic metrics are shown in Table 6, and those of acoustic metrics are shown in Table 7

| Dataset | Model | Pitch mean | Pitch STD | Energy mean | Energy STD | HTN ratio | Speech duration | Speaker similarity |
|---|---|---|---|---|---|---|---|---|
| DailyTalk | Proposed | 0.81 | 0.26 | **0.73** | **0.07** | 0.56 | **0.25** | **0.86** |
| | − *style in context* | 0.82 | 0.27 | 0.72 | 0.06 | 0.58 | 0.13 | 0.85 |
| | − *audio input* | 0.81 | 0.28 | 0.70 | 0.05 | **0.60** | 0.17 | **0.86** |
| | + *ASR & style* | **0.84** | **0.33** | 0.70 | 0.06 | 0.57 | 0.22 | 0.85 |
| PodcastFillers | Proposed | 0.68 | 0.28 | 0.82 | **0.58** | **0.51** | 0.20 | **0.84** |
| | − *style in context* | 0.66 | 0.26 | 0.82 | 0.43 | 0.48 | 0.19 | 0.83 |
| | − *audio input* | 0.70 | 0.27 | **0.83** | 0.49 | 0.48 | 0.22 | 0.82 |
| | + *ASR & style* | **0.75** | **0.28** | 0.82 | 0.48 | 0.49 | **0.28** | **0.84** |

Table 7: Acoustic evaluation results for the proposed and ablated models for ablation study. The results for the proposed model are copied from Table 3.

## C  Subjective Evaluation Details

To ensure high-quality evaluation from MTurk, we followed Li et al. (2024) by enabling the following filters:

- HIT Approval Rate (%) for all Requesters' HITS: `greater than 95`.
- Location: `is UNITED STATES (US)`.
- Number of HITs Approved: `greater than 50`.

We provided the following instructions for rating the naturalness and coherence:

- Naturalness:

      ```
      Rate how natural it is like being produced by human, compared to
      a computer or robot, evaluate based on the content spoken and
      speech quality.
      ```

- Coherence:

      ```
      Rate how coherent the response is to the previous conversation,
      e.g., whether the person speaking is in the same emotion or the
      same person, or the response is relevant to previous
      conversation.
      ```

Additionally, for the evaluation of the DailyTalk dataset, we specifically informed the raters that the speakers are not native speakers and should be taken into consideration while rating naturalness.

An example survey used for our subjective evaluation can be found at `https://survey.alchemer.com/s3/7782786/this-colm2024-b2`. Each rating was paid $5 for completing this 15-minute survey.

