# OpenReview forum: "StyleTalker: Finetuning Audio Language Model and Style-Based Text-to-Speech Model for Fast Spoken Dialogue Generation"
_colmweb.org/COLM/2024/Conference — COLM_

### Official Review · Reviewer_ENYL · 2024-05-07

**Rating:** 7
**Confidence:** 4
**Ethics Flag:** 2

**Summary:**

The paper presents StyleTalker, an audio language model (QwenLM)) finetuned to encode the speech and style inputs as well as the dialogue context and that generates as output the text and style of the other speaker. The output of the QwenLM is the input of a finetuned StyleTTS which generates the speech output with the given style. Authors compare this pipeline QwenLM+StyleTTs, namely StyleTalker to two baselines, a classical cascade pipeline (Whisper +  Qwen-7B + StyleTTS2) and an end-to-end approach SpeechGPT on two datasets DailyTalk and PodcastFillers (with more spontaneous conversations). Results show that the proposed StyleTalker performs better according to semantic and acoustic metrics as well as according to a human evaluation. However, compared to the cascade approach it has a larger word error rate on PodcastFillers. Regarding the real-time factor, the proposed model is faster than baselines. An ablation study shows the positive impact of style in context.

**Ethics Concerns Details:**

Human Evaluation was conducted but little is said about how evaluators were rewarded.

**Questions To Authors:**

-Why an ASR is used to process the output of the TTS? Would it be possible to create an overlapping between recognizing what the system has just said and what the user is currently saying? How was this handled in the proposed implementation?
-How much were evaluators  paid?
-Please remove the spaces between text and footnotes.

**Reasons To Accept:**

Authors compare their proposed pipeline QwenLM+StyleTTs, namely StyleTalker against two baselines, a classical cascade pipeline (Whisper +  Qwen-7B + StyleTTS2) and an end-to-end approach SpeechGPT on two datasets DailyTalk and PodcastFillers (spontaneous conversations). Results show that the proposed approach perform better according to semantic and acoustic metrics. It also performs better with regards to a human evaluation in terms of coherence and naturalness. However, compared to the cascade approach it has a larger word error rate on PodcastFillers. Regarding the real-time factor, the proposed model is faster than baselines. An ablation study shows the positive impact of style in context.

**Reasons To Reject:**

Authors mention very fast in the paper that they do use an ASR to process the system's outputs? Would it be possible to create an overlapping between recognizing what the system has just said and what the user is currently saying? How was this handled in the proposed implementation?
Ethical concerns: how much were evaluators  paid?
The proposed model has a larger word error rate on PodcastFillers compared to the cascade baseline.

---

> ### Author Rebuttal · Authors · 2024-05-31
>
> We thank Reviewer ENYL for the detailed review and constructive feedback on our paper. We appreciate the recognition of our work's strengths and the insightful questions raised. We would like to address the concerns and provide clarifications on certain points:
>
> 1. **Use of ASR for Transcribing Texts:**
>
> The ASR is used to transcribe the texts as context for future conversation turns. This is necessary because using the original audio as input would be slower due to the higher number of audio tokens compared to text and style tokens. We will make this clearer in the revised version of our paper to emphasize the efficiency gained by using text transcriptions in our system.
>
> 2. **Handling Overlapping (Turn Taking):**
>
> We assume the reviewer is referring to "turn-taking" or interruption in spoken dialogue systems (SDS). In our current work, we do not explicitly model turn-taking or interruptions. We acknowledge this as a limitation and will discuss it in the conclusion section. We will propose feasible approaches to address this, such as using a causal audio encoder to predict whether the speaker has finished speaking. This will provide a direction for future improvements in handling conversational dynamics more effectively.
>
> 3. **Evaluator Payment Information:**
>
> We will include detailed information about the payment to the evaluators in the appendix in the revised version. This addition will address the ethical concerns raised and ensure transparency in our evaluation process.

---

> > ### Comment · Reviewer_ENYL · 2024-06-04
> > **Thank you for your answers.**
> >
> > Authors addressed my concerns in their response as they promise to explain better in the paper: (i) why do they use ASR for Transcribing Texts, (i)the turn taking limitation and (iii) they will include detailed information about the payment to the evaluators in the appendix.

---

### Official Review · Reviewer_N9VV · 2024-05-10

**Rating:** 7
**Confidence:** 3
**Ethics Flag:** 1

**Summary:**

The authors describe and evaluate an architecture for a speech conversational LM based on finetuning an audio+text LM and a style-aware TTS model.

**Questions To Authors:**

What are the links between the authors' ideas developed in this paper and the translation-oriented expressivity-aware models developed by META (SEAMLESS / SEAMLESS-Expressive)?

**Reasons To Accept:**

- the overall architecture is novel (as far as I can tell) and smart (I liked the idea of performing speech response generation and current input ASR in parallel)
- the authors provide a demo, and it seems that the source code is planned to be released when the paper is published

**Reasons To Reject:**

- the description of the model could have been clearer. For instance, the way the "style vector" is defined/obtained is described in section 3.1.1, which makes its use in the overall description of the model in section 3.1 quite difficult to understand (on a side note, it seems to me that this style vector is similar to that of the Translatotron or that of META's SONAR EXPRESSIVE paper, none of which are cited)
- the name of the model introduced by the authors is StyleTalker. There already exists a model with this exact same name (arXiv paper: https://arxiv.org/pdf/2208.10922). The title of the paper is clear about what this other StyleTalker is: a "One-shot Style-based Audio-driven Talking Head Video Generation" model. It is thematically close enough (yet different) to this submission, so that the authors should use another name for their model, and should have checked in advance whether the name "StyleTalker" was indeed available for them to use.

---

> ### Author Rebuttal · Authors · 2024-05-31
>
> We thank Reviewer N9VV for the detailed review and the constructive feedback on our paper. We appreciate the recognition of the novelty and design of our architecture and the positive remarks about our demo and planned code release. We would like to address the concerns raised and provide clarifications on certain points:
>
> 1. **Clarification of the Style Vector:**
>
> We acknowledge that the description of the style vector could be clearer. We will clarify the definition and method of obtaining the style vector in Section 3.1, before Section 3.1.1. Additionally, we will include a comparison to SEAMLESS to highlight the differences and similarities between our approach and the translation-oriented expressivity-aware models developed by META.
>
> 2. **Name Adjustment for StyleTalker:**
>
> Thank you for letting us know about this work. We were unaware of it. We understand the concern regarding the name "StyleTalker" and the potential confusion with the existing model of the same name. To address this, we will add a dash between "style" and "talker," renaming our model to "Style-Talker." This adjustment will differentiate our work and avoid any thematic overlap with the existing model.

---

> > ### Comment · Reviewer_N9VV · 2024-06-05
> >
> > Thank you very much for your response. Regarding point 1, please make sure you cite all relevant literature (not only the Seamless paper). Regarding point 2, I am sincerely not convinced that adding a dash is enough to differentiate the models. Too often do we see typographic variation in how model/architecture/corpus names are written (e.g. open-flamingo / OpenFlamingo / Open flamingo / Open-Flamingo etc.), and a dash is not enough to ensure disambiguation. I strongly suggest coining a whole new name.
> >
> > Overall, I see no reasons to modify my score.

---

### Official Review · Reviewer_VCtQ · 2024-05-15

**Rating:** 6
**Confidence:** 3
**Ethics Flag:** 1

**Summary:**

The paper presents a spoken dialog system (SDS) consisting of an audio LLM (Qwen-Audio) followed by a TTS model (StyleTTS 2). Using an audio LLM avoids the explicit ASR transcription step of a conventional cascaded SDS, reducing latency. The audio LLM is fine-tuned to generate both the response text and response prosodic features -- two inputs to the TTS model. Prosodic features are learned embeddings of special tokens. As Qwen-Audio does not seem to handle multiple audio inputs, the input utterance is ASRed and processed by a prosody encoder at response playback time, to become part of the LLM input (text / prosodic features) at the next turn. The system outperforms both cascaded and E2E baselines at 2x speedup.

**Questions To Authors:**

* Prosodic / acoustic styles are not introduced in the TTS model section, yet referred to in subsequent sections.

* The caption of Table 2: "F-1 score (%)" appears out of place there.

**Reasons To Accept:**

* The proposed system is a simple yet effective improvement of a cascaded SDS.
* The paper presents strong evaluation results (naturalness, dialog coherence) at a significant speedup.
* The paper is clearly written.

**Reasons To Reject:**

* While ingenious, the proposed approach appears somewhat narrow and relevant only when using an audio LLM that does not handle multiple audio inputs. The paper should make this clear.

* The paper is missing a discussion and experiments on working around the seemingly redundant ASR / prosody encoder calls at playback time. How does the proposed approach compare to using an audio LLM that natively handles multiple audio inputs? (Does Qwen-Audio-Chat from the Qwen-Audio paper handle multiple audio inputs?) If we stick with Qwen-Audio, can we reuse the LLM's embedding of the utterance to compute the prosodic features for the next turn? And fine-tune Qwen-Audio to do both response generation and ASR in one go?

* The presentation can be improved. The paper appears too verbose. The introduction and related work section overlap significantly and should be simplified (e.g. the related work section should only highlight the work the paper directly compares to). Parts of the appendix (A.1 on how style is encoded in the LLM and A.3 on the baselines) should probably be in the main paper.

* I'm a bit concerned if the paper is a good [fit for COLM](https://colmweb.org/cfp.html) as it is more about an application of an LLM, rather then the science of large language modeling.

---

> ### Author Rebuttal · Authors · 2024-05-31
>
> We thank the reviewer for the constructive feedback and the detailed evaluation of our work. We would like to address the concerns raised and provide clarifications on certain points:
>
> 1. **Handling Multiple Audio Inputs & Discussions:**
>
> The audio LLM in our system is indeed capable of handling multiple audio inputs. However, doing so significantly slows the processing speed due to the higher number of audio tokens compared to text tokens and a style vector. In addition, it does not provide substantial improvements when the input consists of previous turns. For the same efficiency and performance reasons, we did not reuse the LLM’s embedding from previous turns, even though it is technically possible. Moreover, these previous turns can be efficiently summarized using their transcription texts and a style vector. We will expand on this discussion in the camera-ready version to provide a clearer understanding of the trade-offs involved.
>
>
> 2. **Simplifying Introduction and Related Work:**
>
> We appreciate the suggestion to streamline the introduction and related work sections. We will simplify these sections and integrate relevant information from Appendix A.1 and A.3 into the main paper to improve the clarity and coherence of our presentation.
>
> 3. **Category of Submission:**
>
> Our paper was submitted under the "novel applications in LLM" category rather than the science of LLMs. We believe our work aligns well with the topics of COLM, as it demonstrates an innovative application of LLMs in spoken dialog systems. We will make this distinction clearer in the revised submission.
>
> 4. **Other Minor Issues:**
>
> We will add more detailed descriptions of the acoustic and prosodic styles used in StyleTTS 2 in the revision. We will correct the misplaced percentage sign in the caption of Table 2 to ensure accuracy and clarity in our presentation.

---

> > ### Comment · Reviewer_VCtQ · 2024-06-07
> > **Response to authors**
> >
> > Thank you for your response.
> >
> > I keep my score unchanged. I believe the paper is good, but it's quite not there yet.
> >
> > My rejection points 1 and 2 still hold and are not addressed sufficiently in the rebuttal, which in fact raises more questions (e.g. it is unclear why handling multiple audio inputs is slower -- unless the authors refer to a naive baseline in which all input context is re-encoded entirely at each turn, without any state sharing). I believe these are issues at the core of the paper and, even as the rebuttal shows, if addressed properly, would require substantial changes to the paper, including new experiments.

---

### Official Review · Reviewer_mnFk · 2024-05-24

**Rating:** 6
**Confidence:** 4
**Ethics Flag:** 1

**Summary:**

The authors propose a novel framework combining audio LLM and style based TTS for the task of spoken dialog generation. The proposed system significantly outperforms SpeechGPT and Cascaded system. The demo indicates that the audio quality generated from StyleTalker sounds quite natural on the DailyTalk corpus. The paper is clear in general and includes detailed analysis of the results and evaluation.

**Reasons To Accept:**

1.) The paper proposes a new framework for spoken dialogue generation using a combination of AudioLLM and style based TTS.
2.) The quality of synthesized audio for the DailyTalk dataset is quite natural sounding.
3.) Comparison between the end-to-end system i.e. SpeechGPT, Cascaded framework (ASR + LLM + TTS) and StyleTalker shows StyleTalker's superiority in terms of the quality of audio generated as well as processing speeds.
4.) The authors present a comprehensive analysis of results comparing different models in terms of processing times, quality of audio synthesized on 2 datasets, with and without inclusion of style information, ASR transcripts, etc. making it a valuable contribution and a great read.

**Reasons To Reject:**

1.) For the PodcastFillers data, the synthesized audio quality is still not very good.
2.) Table 5 shows that using ASR transcripts (against audioLLMs) can lead to better performance but slower processing time.
3.) Response tome using StyleTalker may still not surpass an SDS with real time as the authors cite as well.

---

> ### Author Rebuttal · Authors · 2024-05-31
>
> We appreciate Reviewer mnFk's detailed review and valuable feedback on our paper. We would like to address the concerns raised and provide clarifications on certain points:
>
>
> 1. **Audio Quality on PodcastFillers Dataset:**
>
> The audio quality depends on the pre-trained TTS model StyleTTS 2, which is the current public state of the art in TTS. We did not propose a new synthesis model in this work but fine-tuned pre-trained models. The current TTS models, including StyleTTS 2, may face challenges when working with in-the-wild datasets like PodcastFillers. We acknowledge this limitation and will include it as a future research direction for TTS communities on improving TTS models with such diverse datasets. This clarification will be added to the conclusion section of the camera-ready version.
>
> 2. **Performance Comparison with ASR Transcription:**
>
> The performance with ASR transcription is on par with the proposed system without transcription, both achieving acoustic aggregated scores of 1.07. Additionally, our proposed system without ASR transcription achieves a slightly higher semantic score. Thus, ASR transcription is not necessary for our system, as evidenced by these results. This finding highlights the efficiency and effectiveness of our approach without the need for additional ASR processing. We will make this point more clear in the camera-ready version.
>
> 3. **Real-Time Response Using StyleTalker:**
>
> We acknowledge the concern that although our model has surpassed the traditional cascade pipeline, the response time of StyleTalker may not surpass an SDS with real-time streaming ASR. We have mentioned this in the conclusion section and will expand further on this point in the camera-ready version.

---

> > ### Comment · Reviewer_mnFk · 2024-06-06
> > **Thank you for your response**
> >
> > The authors mention that they will clarify the points in the camera ready version of the paper. I do not see a reason to modify the score further as I am leaning towards accepting the paper in my original review.

---

### Decision · Program_Chairs · 2024-07-10

**Decision:**

Accept

**Comment:**

StyleTalker makes a simple and clever improvement over the classic ASR->LLM->TTS pipeline that many audio dialogue systems use, by joining together the ASR and LLM systems into a single audio LLM that produces both text and style simultaneously.

Overall, reviewers find StyleTalker worthy of publication for a few key reasons:
- The proposed strategy is novel, yet simple (which is a virtue!)
- The dialogue produced by the system is both natural and faster than baselines, moving past the current Pareto frontier of this tradeoff.
- The paper presents deep analysis of the underlying variables that affect the proposed approach, e.g., a thorough ablation that consider the inclusion/exclusion of style information, audio input, etc.
- The code and model will be open sourced.

The authors have agreed to make a few points clearer, mainly:
- The definition and methodology around the style vector.
- The use of ASR for encoding future turns.
- The feasibility of handling multiple inputs.
- How evaluators were compensated.

Reviewers note that there is room for improvement, especially agreeing on:
- It is not clear that StyleTalker will surpass all SDS systems with more efficient ASR systems.
- Performance on in-the-wild applications, such as those represented by the PodcastFillers dataset, are somewhat disappointing.

Overall, however, the paper makes clear and well-presented contributions, and is recommended for acceptance.